# Protocol for the development of a core outcome set for neonatal sepsis (NESCOS)

Petek Eylul Taneri[1,2]*, Jamie J. Kirkham[3], Eleanor J. Molloy[4,5], Linda Biesty[2,6], Richard A. Polin[7], James L. Wynn[8], Barbara J. Stoll[9,10,11], Niranjan Kissoon[12], Kondwani Kawaza[13], Mandy Daly[14], Aoife Branagan[4,5], Lívia Nagy Bonnard[15], Eric Giannoni[16], Tobias Strunk[17], Magdalena Ohaja[2], Kenneth Mugabe[18], Denise Suguitani[19], Fiona Quirke[1,20], Declan Devane[1,2,6]

1 HRB-Trials Methodology Research Network, University of Galway, Galway, Ireland, 2 School of Nursing and Midwifery, University of Galway, Galway, Ireland, 3 Centre for Biostatistics, Manchester Academic Health Science Centre, The University of Manchester, Manchester, United Kingdom, 4 Department of Neonatology, The Coombe Hospital, Dublin, Ireland, 5 Department of Paediatrics and Child Health, Trinity College Dublin, Trinity Research in Childhood Centre (TRiCC), Neonatology, Children's Health Ireland, Dublin, Ireland, 6 Evidence Synthesis Ireland & Cochrane Ireland, University of Galway, Galway, Ireland, 7 Department of Paediatrics, College of Physicians and Surgeons, Columbia University, New York, New York, United States of America, 8 Department of Paediatrics, College of Medicine, University of Florida, Gainesville, Florida, United States of America, 9 China Medical Board, China, 10 Emory University School of Medicine, Atlanta, GA, United States of America, 11 McGovern Medical School of the University of Texas Health Science Center at Houston, Houston, Texas, United States of America, 12 Department of Paediatrics, College of Medicine, University of British Columbia, Vancouver, Canada, 13 Department of Paediatrics and Child Health, Kamuzu University of Health Sciences, Blantyre, Malawi, 14 Education and Research, Irish Neonatal Health Alliance, Bray, Ireland, 15 Patient Advocacy, Melletett a helyem Egyesület, Budapest, Hungary, 16 Clinic of Neonatology, Department Mother-Woman-Child, Lausanne University Hospital and University of Lausanne, Lausanne, Switzerland, 17 Neonatal Directorate, Child and Adolescent Health Service, Wesfarmers' Centre for Vaccines and Infectious Diseases, Telethon Kids Institute, University of Western Australia, Perth, Australia, 18 Mbale Regional Referral Hospital, Busitema University Faculty of Health Sciences, Mbale, Uganda, 19 Brazilian Parents of Preemies' Association, Brazil, 20 School of Medicine, University of Limerick, Limerick, Ireland

* petekeylul.taneri@universityofgalway.ie

**Data Availability Statement:** No datasets were generated or analysed during the current study. All relevant data from this study will be made available upon study completion.

## Abstract

Neonatal sepsis is a serious public health problem; however, there is substantial heterogeneity in the outcomes measured and reported in research evaluating the effectiveness of the treatments. Therefore, we aim to develop a Core Outcome Set (COS) for studies evaluating the effectiveness of treatments for neonatal sepsis. Since a systematic review of key outcomes from randomised trials of therapeutic interventions in neonatal sepsis was published recently, we will complement this with a qualitative systematic review of the key outcomes of neonatal sepsis identified by parents, other family members, parent representatives, healthcare providers, policymakers, and researchers. We will interpret the outcomes of both studies using a previously established framework. Stakeholders across three different groups i.e., (1) researchers, (2) healthcare providers, and (3) patients' parents/family members and parent representatives will rate the importance of the outcomes in an online Real-Time Delphi Survey. Afterwards, consensus meetings will be held to agree on the final COS through online discussions with key stakeholders. This COS is expected to minimize outcome heterogeneity in measurements and publications, improve comparability and synthesis, and decrease research waste.

**Funding:** This research was funded by the Health Research Board (HRB, Ireland) through funding to the HRB Irish Network for Children's Clinical Trials (In4Kids) [Grant number: CTN-2021-007] Funders did not play any role in the study design, data collection (when applicable) and analysis, decision to publish, or preparation of the manuscript.

**Competing interests:** The authors have declared that no competing interests exist.

## Introduction

Sepsis is a major global public health problem. Children account for more than half of all sepsis cases, most of whom are neonates [1]. The global incidence of neonatal sepsis has been estimated at 2824 (95% CI 1892 to 4194) cases per 100,000 live births, with a mortality rate of 17.6% (95% CI 10.3% to 28.6%) and a higher burden in low-middle income countries [2]. In addition to the increased risk of early death, neonatal sepsis may cause long-term neurodevelopmental delay [3] and neurodevelopmental impairments such as cerebral palsy and neurosensory deficits [4]. Neonatal sepsis is also associated with impaired physical growth, increased risk of bronchopulmonary dysplasia and multiple organ failure [3].

There is significant variation in definitions of neonatal sepsis [5], clinical signs of neonatal sepsis [6], and antibiotic use [7]. The lack of a universally accepted definition for neonatal sepsis inhibits efforts to improve accurate diagnostic and prognostic testing methods for this vulnerable population [8]. In addition, there is substantial heterogeneity in the outcomes measured and reported in studies evaluating the effectiveness of treatments for neonatal sepsis, with Henry et al. identifying 88 outcomes across 90 studies [9]. This heterogeneity in outcomes limits the ability to compare, contrast, and synthesise [10] the findings of individual studies and contributes to research waste.

Furthermore, the challenges experienced by those who use research to inform decisions about their health care are exacerbated further by outcome-reporting bias [11]. Outcomes in which the intervention has a statistically significant effect are more likely to be fully reported [12]. Developing a Core Outcome Set (COS) for treatments for neonatal sepsis could help minimise these current shortcomings in reporting challenges.

A COS is 'the minimum that should be measured and reported in all clinical trials of a specific condition and could also be suitable for use in other types of research and clinical audit' [13]. The COS should always be measured, collected, and reported. Researchers may measure and report outcomes additional to the COS, but the COS should be reported in its entirety [11]. The advantages of the widespread use of a COS includes increased outcome consistency across trials, a likely decrease in selective reporting, and increased opportunity for a study to contribute to syntheses across outcomes included in the COS [14]. Importantly, using a COS helps ensure that the outcomes measured and reported are those that matter to stakeholders, including patients and/or their carers [15].

This study aims to develop a COS for studies evaluating the effectiveness of treatments for neonatal sepsis.

## Materials and methods

### Overview

The University of Galway, Research Ethics Committee provided ethical approval for this project (Reference Number: 2022.10.002). During the Delphi survey and consensus meeting phases, participants were provided with detailed project information via participant information leaflets. They were then cordially invited to partake in the study, contingent upon their agreement as evidenced by completing the standardized consent forms. These forms, which included a thorough briefing, were utilized to obtain written consent from the participants. Integral to our adherence to ethical standards, these documents were incorporated into our submission to the ethical committee and were subsequently granted approval.

This protocol has been prepared according to COS-STAP Statement recommendations [16]. We are not aware of any published COS evaluating the effectiveness of interventions for treating neonatal sepsis. The protocol of this study has been registered with the Core Outcome

**Fig 1. Schematic of COS development.**

Measures in Effectiveness Trials (COMET) initiative (https://www.comet-initiative.org/Studies/Details/2118).

This study will be conducted in four stages (Fig 1):

1. Qualitative systematic review

2. Delphi survey

3. Consensus meetings

4. Dissemination and implementation

## Stage 1: Qualitative systematic review

Recently, Henry et al. published a systematic review of core outcomes from randomised trials of therapeutic interventions in neonatal sepsis. They reported 88 unique outcomes from 90 included studies [9].

The outcomes chosen and reported by researchers or clinicians alone may not be important and/or relevant for other stakeholders, such as patients, caregivers, or other decision-makers [17]. The outcomes considered during the consensus process should represent the views of all relevant stakeholders. Therefore, it has been advised that COS developers consider different approaches to establishing an initial list of outcomes for prioritisation, such as collecting data from patient interviews or analysing qualitative research studies focusing on patients' opinions [18]. Based on this recommendation, several qualitative systematic reviews have been published to support the COS development processes in, for example, female chronic pelvic pain [17], type 2 diabetes [18], and neonatal care [19].

Our proposed qualitative systematic review was registered with The International Prospective Register of Systematic Reviews (PROSPERO ID: CRD42022344485). The review will be conducted and reported following the guidelines of the ENTREQ statement and Preferred Reporting Items for Systematic Reviews and Meta-Analyses (PRISMA) [19, 20].

Review/synthesis question: What are the key outcomes of neonatal sepsis identified by parents, other family members, parent representatives, healthcare providers, policymakers, and researchers that should be included in a 'long list' of outcomes in a Delphi Survey as part of the development of a COS on treatments for neonatal sepsis?

## Inclusion criteria

The PerSPEcTiF framework was used to structure the description of inclusion criteria [21] as per qualitative review guidelines [22, 23] (Table 1).

We will include studies with a qualitative study design such as ethnographic, grounded theory, historical, and case study; those that used qualitative data-collection methods, including observations, textual or visual analysis, and interviews (individual or group). We will also include mixed-method studies with a qualitative component where participant experiences are reported separately. We will exclude abstracts, randomised trials, clinical trials, intervention studies, cross-sectional studies, case-control studies, prospective and retrospective cohort studies, letters to the editor, conference proceedings, and systematic reviews with or without meta-analyses.

**Search strategy.** With no date restrictions, a comprehensive literature search will be carried out using MEDLINE, EMBASE, CINAHL, and PsycINFO databases. The complete search strategy is presented in S1 File.

The reference lists of included studies will be hand-searched to identify other potentially relevant studies [24–27]. Findings of the literature search will be transferred to Endnote, and after deduplication, abstract and title screening will be done via Rayyan [28].

**Study selection.** The titles and abstracts will be screened by two reviewers independently using the eligibility criteria, and discrepancies will be resolved through discussion with a third review author where necessary. For all studies that appear to fulfil the eligibility criteria based on the abstract, two authors will examine full-text articles independently. Discrepancies will be resolved again through discussion with a third review author where necessary.

**Quality assessment.** Two reviewers will independently assess the quality of the included studies using the Critical Appraisal Skills Programme (CASP) tool [29]. While studies will not be excluded based on this assessment of methodological limitations, the assessment will be reported in the final write-up of our systematic review. The results will be reported in the manuscript to inform readers about the methodological quality of the included studies.

**Data extraction.** One review author will extract the data using a pre-piloted data extraction form to obtain study characteristics from qualifying studies. One additional review author will double-check the data extraction performed by the first review author and verify that all relevant data are extracted. Author information, publication year, study design, number of participants, data collection and analysis methods, and text excerpts relevant to outcomes will be noted on the form.

**Table 1. The PerSPEcTiF question formulation framework.**

| Per | S | P | E | (C) | Ti | F |
|---|---|---|---|---|---|---|
| Perspective | Setting | Phenomenon of interest/ Problem | Environment | Comparison (optional) | Time/ Timing | Findings |
| Parents, other family members, parent representatives, healthcare providers, policymakers, researchers | Any setting | The outcomes of treatments for neonatal sepsis are important to parents, other family members, parent representatives, healthcare providers, policymakers, and researchers that should be included in a 'long list' of outcomes contributing to the COS | High, middle and low-income countries | - | Anytime following diagnosis of sepsis | All outcomes regarding the phenomenon of interest |

**Data analysis.** Drawing on the principles of thematic synthesis [30], extracted data related to outcomes will be coded line by line, and similarities will be identified. Codes will be grouped to define distinct, descriptive themes related to outcomes of therapeutic interventions in neonatal sepsis. Two reviewers will review and interpret the outcomes noted in the themes and will map these to a framework of outcome domains presented by Webbe et. al's "Core outcomes in neonatology: development of a core outcome set for neonatal research." [31]. The following domain headings will be used to guide this mapping: survival, respiratory, cardiovascular, gastrointestinal, neurological, genitourinary, skin, surgical, development (gross motor/fine motor/cognitive/special senses/speech and social), psychosocial, healthcare utilisation, outcomes related to parents, outcomes related to healthcare workers, general outcomes and miscellaneous. If the identified outcomes do not map to any domains, the reviewers will develop additional relevant domains.

The outcomes determined at the end of this phase will be brought forward to Stage 2.

## Stage 2: Delphi survey

Delphi studies were used first in the 1960s by Dalkey and Helmer of the Rand Corporation to obtain consensus on the views of a group of experts [32]. Traditionally, the Delphi procedure helps develops consensus among experts by conducting controlled feedback on the findings of multiple survey rounds (usually) or interviews [33]. It can foster collective ownership and promote unity among people with different points of view [34]. The traditional Delphi technique is used extensively in health care to achieve consensus on various topics such as mental health [35–37], cancer [38, 39], and primary health care [40, 41].

The traditional Delphi technique's recurring and multiple feedback nature necessitates a significant amount of time, which increases the risk of participant attrition across rounds [42]. In 2006, Gordon & Pease introduced the Real-Time Delphi (RTD) as an alternative to multi-round Delphi surveys, aiming to improve the process's efficiency and shorten the required time. In this method, participants can view the group's responses for each question in real time and the total number of responses simultaneously [43]. RTD is round-less; however, in light of changing feedback, participants can be recommended to revisit and re-rate questions. The responses participants will see on visiting an RTD survey reflect those who have participated in the survey up to that point rather than reflecting all survey respondents after each round as in traditional Delphi.

The number of papers using RTD is increasing. The RTD has been used to develop consensus across a range of settings, including economics, [43, 44] transportation [44], education [45, 46], religion [47], professional development [48], and sustainability [49]. RTD has been used in healthcare studies that aim to develop a medication adherence technologies repository [50] and determine the negative impacts of substance use disorders among people with HIV in the United States [51]. Several researchers have incorporated an RTD approach into the COS development processes [52, 53].

A comparison of real-time and traditional Delphi approaches for the future of the logistics industry in the year 2025 showed no statistically significant differences in participants' projections on the future of logistics between the two studies [42]. A study comparing online Delphi and Real-Time Delphi surveys suggested that the RTD may reduce survey time and expenses and increase tester convergence and consensus. However, the study is small (n = 12) and lacks clarity on experimental methods [54]. A randomised trial comparing a three-round with a Real-Time Delphi approach on outcomes prioritised as part of a core outcome set development is currently awaiting reporting [52].

**Participants.** We aim to include at least 100 stakeholders across three different stakeholder groups i.e., (1) researchers, (2) healthcare providers, and (3) patients' parents/family

members and parent representatives. We plan to include participants from high, middle, and low-income countries. Participants will be sought through professional associations and groups representing different stakeholders. Through the encouragement of the participants to forward the invitation, especially to stakeholders from low-income countries, snowball sampling will also be encouraged.

**Procedures.** Previously, 88 outcomes were identified from 90 randomised trials [9]. Those will be mapped to Webbe et al.'s framework [31]. If a domain and/or outcome cannot be mapped, a new domain/outcome will be created. This will be combined with the domains and outcomes identified in Stage 1. Duplicate domains and outcomes will be removed, leaving a list of unique domains and outcomes for presentation as the 'long list' outcomes for use in the Delphi survey.

We will develop a survey in which participants will be invited to rate the importance of each outcome. Participants will rate the importance of each outcome for inclusion in the COS on a 9-point Likert scale (i.e., 1–3 limited importance, 4–6 important but not critical, and 7–9 critical) [11].

Once a participant rates an outcome, they will be presented with 'real-time' feedback on the portion of participants from each stakeholder group and all participants' overall rating for each point on the 9-point scale for each outcome. Participants will then consider their ratings based on real-time feedback on responses from other participant groups and the group overall.

Before the survey goes live, we want to ensure that the survey is populated with feedback responses representative of the stakeholder groups. For this, we will recruit at least 5 participants from each stakeholder group (researchers, healthcare providers, patients' parents/family members and parent representatives). These participants will also be included in the final survey numbers. This first group of participants will be invited to revisit the survey when it is live, to see stakeholder responses and amend their rating for each outcome if they wish.

Once the main survey goes live, participants will have a 3-week window during which they will be invited to re-visit the survey and review responses if they wish. They can choose to retain their original rating for each outcome or alter their rating based on how others have responded. Based on a comparison of software-based tools for RTD [55, 56], we will use Surveylet (https://calibrum.com) platform for our RTD.

At the end of the three weeks, outcomes that scored 7–9 by 70%, and 1–3 by less than 15% of participants across all stakeholder groups will be brought forward to the consensus meetings. When 50% or fewer participants score 7–9 in each stakeholder group that outcome will be excluded. This consensus definition was created to ensure that a measure could not achieve consensus if a minority stakeholder group mostly rated it as 'limited importance' and was used previously during the development of other COSs [57–59].

## Stage 3: Consensus meetings

The consensus meetings aim to agree on the final COS through online discussions with key stakeholders.

**Participants.** Consensus meetings will be conducted with an international panel of stakeholders representing patients' parents/family members and parent representatives, researchers, and healthcare providers from high, low, and middle-income countries.

**Schedule.** We will hold two independent online consensus meetings, including at least three people from each stakeholder group, to make it feasible for the participants living in different time zones. The unique outcomes identified for inclusion in the COS from each of the two consensus meetings will be discussed at a third, final online consensus meeting attended by stakeholders representing patients' parents/family members and parent representatives, researchers, and healthcare providers from high, low, and middle-income countries.

During the meetings, the outcomes emerging from the RTD will be presented to the participants, along with the voting patterns of the stakeholder groups for each outcome. An experienced facilitator who will not participate in voting will chair each consensus meeting. There will be anonymous, computerised voting after a discussion of each outcome. If at least 80% of participants, including at least one representative from each stakeholder group, vote in favour of an outcome, it will be included in the COS.

## Stage 4: Dissemination and implementation strategy

Once the consensus meeting is over and the final COS has been approved, a paper will be written, published, and distributed in accordance with the COS-STAP recommendations [16]. The final COS will be made available through the COMET database. We will present our results at national and international conferences to encourage researchers and clinicians to use the COS. We will target the audience/key stakeholders (e.g., all participants in the survey, known maternity and neonatal care researchers, the Cochrane Pregnancy and Childbirth Group, and the Cochrane Neonatal Group.

## Discussion

Despite advancements in the quality of neonatal care, sepsis-related morbidity and mortality are growing concerns, particularly in low-middle-income countries [60]. Currently, there is inconsistency in the reporting of outcomes in studies on treatments of neonatal sepsis. A COS can help standardize outcomes measured and reported in studies and enable comparisons between and synthesis across them [61]. By conducting a systematic review, RTD survey, and consensus meetings, we will develop a COS for studies evaluating the effects of treatments for neonatal sepsis. We anticipate that this COS will minimise heterogeneity in outcomes measured and reported, enable better comparison and synthesis, and reduce research waste.

## Study status

The study is ongoing, in Stage 1 (systematic review).

## Supporting information

**S1 Checklist. PRISMA-P (Preferred Reporting Items for Systematic review and Meta-Analysis Protocols) 2015 checklist: Recommended items to address in a systematic review protocol\*.**
(DOC)

**S1 File. NESCOS search strategy.** The complete search strategy of the qualitative systematic review of the NESCOS project.
(DOCX)

## Author Contributions

**Conceptualization:** Petek Eylul Taneri, Jamie J. Kirkham, Eleanor J. Molloy, Linda Biesty, Richard A. Polin, James L. Wynn, Barbara J. Stoll, Niranjan Kissoon, Kondwani Kawaza, Mandy Daly, Aoife Branagan, Lívia Nagy Bonnard, Eric Giannoni, Tobias Strunk, Magdalena Ohaja, Kenneth Mugabe, Denise Suguitani, Fiona Quirke, Declan Devane.

**Funding acquisition:** Petek Eylul Taneri, Declan Devane.

**Investigation:** Petek Eylul Taneri, Linda Biesty, Declan Devane.

**Methodology:** Petek Eylul Taneri, Jamie J. Kirkham, Eleanor J. Molloy, Linda Biesty, Declan Devane.

**Project administration:** Petek Eylul Taneri, Declan Devane.

**Resources:** Petek Eylul Taneri, Declan Devane.

**Software:** Petek Eylul Taneri, Declan Devane.

**Supervision:** Declan Devane.

**Writing – original draft:** Petek Eylul Taneri, Linda Biesty, Declan Devane.

**Writing – review & editing:** Petek Eylul Taneri, Jamie J. Kirkham, Eleanor J. Molloy, Linda Biesty, Richard A. Polin, James L. Wynn, Barbara J. Stoll, Niranjan Kissoon, Kondwani Kawaza, Mandy Daly, Aoife Branagan, Lívia Nagy Bonnard, Eric Giannoni, Tobias Strunk, Magdalena Ohaja, Kenneth Mugabe, Denise Suguitani, Fiona Quirke, Declan Devane.

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
