## [Editor Report · Decision Letter 0]

21 Aug 2023

PONE-D-23-05983Protocol for the development of a Core Outcome Set for Neonatal Sepsis (NESCOS)PLOS ONE

Dear Dr. Taneri,

Thank you for submitting your manuscript to PLOS ONE. After careful consideration, we feel that it has merit but does not fully meet PLOS ONE’s publication criteria as it currently stands. Therefore, we invite you to submit a revised version of the manuscript that addresses the points raised during the review process.

Given the manuscript reports a robust and well established study design, and the manuscript is well written, as academic editor I am happy to accept your submission without undertaking a formal peer review process. However, you will notice that I am flagging it as a minor revision and I ask that some extremely minor formatting issues are addressed before I can proceed to formally accept the manuscript in the system. For example, I did notice a minor issue - one of the citations (44) in the reference list is incomplete and displayed an error message. I hope you are pleased with this outcome and I look forward to seeing and processing the revised manuscript.

We look forward to receiving your revised manuscript.

Kind regards,

Andrew Harding, PhD

Academic Editor

PLOS ONE
---

## [Author Response · Author response to Decision Letter 0]

30 Aug 2023

Dear editor,

Thank you for your valuable suggestions, which we have now incorporated into the submission. Kindly find below the specific changes we've made as per your recommendations.

Changes we made: We carefully reviewed the PLOS ONE style templates and made the necessary adjustments to ensure that our manuscript meets the criteria set by the journal.

Changes we made: We added captions for the Supporting Information file at the end of the manuscript and revised the in-text citations to align with the provided guidelines. We took into account the guidelines while making these updates.

Changes we made: We thoroughly reviewed the reference list to confirm its completeness and accuracy. There are no instances of retracted papers within the list. It's important to note that we made no additions or removals of references; our sole focus was on rectifying the reference list itself.

Kind regards,

Petek Eylul Taneri

---

## [Editor Report · Decision Letter 1]

10 Nov 2023

PONE-D-23-05983R1Protocol for the development of a Core Outcome Set for Neonatal Sepsis (NESCOS)PLOS ONE

Dear Dr. Taneri,

Thank you for submitting your manuscript to PLOS ONE. After careful consideration, we feel that it has merit but does not fully meet PLOS ONE’s publication criteria as it currently stands. Therefore, we invite you to submit a revised version of the manuscript that addresses the points raised during the review process.

During our internal assessment of the manuscript, we noted that the consent statement is missing. Please provide additional details regarding participant consent. In the ethics statement in the Methods and online submission information, please ensure that you have specified (1) whether consent will be informed and (2) what type you will obtain (for instance, written or verbal, and if verbal, how it will be documented and witnessed). If the need for consent was waived by the ethics committee, please include this information.

We look forward to receiving your revised manuscript.

Kind regards,

Johanna Pruller, PhD

Associate Editor

PLOS ONE

on behalf of

Andrew Harding, PhD

Academic Editor

PLOS ONE
---

## [Author Response · Author response to Decision Letter 1]

14 Nov 2023

Response to Reviewers

Thank you for your valuable suggestions.

1. During our internal assessment of the manuscript, we noted that the consent statement is missing. Please provide additional details regarding participant consent. In the ethics statement in the Methods and online submission information, please ensure that you have specified (1) whether consent will be informed and (2) what type you will obtain (for instance, written or verbal, and if verbal, how it will be documented and witnessed). If the need for consent was waived by the ethics committee, please include this information.

Changes we made: We carefully revised our submission, and we added this information to our manuscript and online submission information:

“During the Delphi survey and consensus meeting phases, participants were provided with detailed project information via participant information leaflets. They were then cordially invited to partake in the study, contingent upon their agreement as evidenced by completing the standardized consent forms. These forms, which included a thorough briefing, were utilized to obtain written consent from the participants. Integral to our adherence to ethical standards, these documents were incorporated into our submission to the ethical committee and were subsequently granted approval.”

---

## [Editor Report · Decision Letter 2]

21 Nov 2023

Protocol for the development of a Core Outcome Set for Neonatal Sepsis (NESCOS)

PONE-D-23-05983R2

Dear Dr. Taneri,

We’re pleased to inform you that your manuscript has been judged scientifically suitable for publication and will be formally accepted for publication once it meets all outstanding technical requirements.

Kind regards,

Andrew Harding, PhD

Academic Editor

PLOS ONE

Additional Editor Comments (optional):

Thank you for the minor changes to your protocol, and we can now accept your manuscript for publication. Good luck with publishing the rest of the work associated with your project.
---

## [Editor Report · Acceptance letter]

24 Nov 2023

PONE-D-23-05983R2 

Protocol for the development of a core outcome set for neonatal sepsis (NESCOS) 

Dear Dr. Taneri:

I'm pleased to inform you that your manuscript has been deemed suitable for publication in PLOS ONE. Congratulations! Your manuscript is now with our production department. 

Kind regards, 

on behalf of

Dr. Andrew Harding 

Academic Editor

PLOS ONE